# Performing arts as a health resource? An umbrella review of the health impacts of music and dance participation

J. Matt McCrary [1,2]*, Emma Redding[3], Eckart Altenmüller[1]

**1** Institute for Music Physiology and Musicians' Medicine, Hannover University for Music, Drama and Media, Hannover, Germany, **2** Prince of Wales Clinical School, Faculty of Medicine, University of New South Wales, Sydney, Australia, **3** Division of Dance Science, Faculty of Dance, Trinity Laban Conservatoire of Music and Dance, London, United Kingdom

\* j.matt.mccrary@gmail.com

**Data Availability Statement:** All relevant data are within the manuscript and its Supporting Information files.

**Funding:** JMM was supported by a Postdoctoral Fellowship from the Alexander von Humboldt

## Abstract

An increasing body of evidence notes the health benefits of arts engagement and participation. However, specific health effects and optimal modes and 'doses' of arts participation remain unclear, limiting evidence-based recommendations and prescriptions. The performing arts are the most popular form of arts participation, presenting substantial scope for established interest to be leveraged into positive health outcomes. Results of a three-component umbrella review (PROSPERO ID #: CRD42020191991) of relevant systematic reviews (33), epidemiologic studies (9) and descriptive studies (87) demonstrate that performing arts participation is broadly health promoting activity. Beneficial effects of performing arts participation were reported in healthy (non-clinical) children, adolescents, adults, and older adults across 17 health domains (9 supported by moderate-high quality evidence (*GRADE criteria*)). Positive health effects were associated with as little as 30 (*acute effects*) to 60 minutes (*sustained weekly participation*) of performing arts participation, with drumming and both expressive (*ballroom*, *social*) and exercise-based (*aerobic dance*, *Zumba*) modes of dance linked to the broadest health benefits. Links between specific health effects and performing arts modes/doses remain unclear and specific conclusions are limited by a still young and disparate evidence base. Further research is necessary, with this umbrella review providing a critical knowledge foundation.

## 1. Introduction

Participation and receptive engagement in the arts are increasingly recognized as being health promoting, most notably in policy briefs [1], a recent World Health Organization-commissioned scoping review [2], and social prescribing initiatives [3]. However, the widespread integration of the arts into healthcare and public health practices is limited by a disparate evidence base; the specifics of the most effective arts interventions–namely the mode (specific 'type' of art–e.g. ballroom dance, singing) and 'dose' (frequency and timing/duration)–for various clinical and public health scenarios are still unclear [4]. Consequently, formulation of evidence-based arts prescriptions and recommendations is presently difficult [4].

Foundation. The funders had no role in study design, data collection and analysis, decision to publish, or preparation of the manuscript.

**Competing interests:** The authors have declared that no competing interests exist.

Participation in the performing arts is the most popular form of arts participation, with up to 40% of EU and US adults participating annually in performing arts activities [5, 6]. Within the performing arts, music and dance participation are the two most popular modes of engagement, both involving engagement with music, and proposed to have common evolutionary origins [7–9]. The health effects of music and dance participation were thus considered likely to be both related and broadly studied and are the focus of this review. 'Performing arts participation' will be used to refer, jointly, to music and dance participation from this point forward.

Performing arts participation is particularly intriguing in a health context in that it combines creative expression with intrinsic levels of physical exertion associated with many health benefits (e.g. moderate–vigorous intensity cardiovascular demands) [10]. Both creative arts and physical activity have been independently linked to broad health benefits [2, 11–13], albeit with a more robust body of evidence supporting the substantial and widespread health impact of physical activity–primary and/or secondary prevention for at least 25 chronic medical conditions and a 9–39% reduction in overall mortality risk [12, 14, 15]. Performing arts and exercise/ physical activity participation are distinguished by a distinctly expressive, rather than exertive, focus of the performing arts; exertion is an intrinsic byproduct, not an objective. Accordingly, the health impact of performing arts participation must be evaluated using frameworks that allow performing arts to remain a primarily expressive activity, but also consider the likely impact of intrinsic physical exertion.

Evidence regarding the full breadth of health impacts of performing arts participation, as well as the modes and doses underpinning these effects, has yet to be compiled, critically appraised and analyzed using a common framework. This umbrella review aims to address this knowledge gap by systematically reviewing and appraising evidence regarding the health effects of performing arts participation, including its impacts on both broad mortality and disease risk and more discrete health-related outcomes, in healthy (non-clinical) adults, adolescents and children. Performing arts participation is hypothesized to have similar health effects as physical activity due to its intrinsic physical exertion, as well as additional effects related to creative expression and engagement with music. Accordingly, a secondary aim of this review is to compile data regarding the intrinsic physical intensity of varying modes of performing arts participation to inform further hypotheses related to relationships between physical intensity and observed effects.

## 2. Methods

### 2.1 Review registration

This review was prospectively registered in the PROSPERO registry (ID: CRD42020191991).

### 2.2 Overview

Following informal literature searches, the authors made an a priori decision that an integrated, three component umbrella review would most effectively address study aims:

1. A systematic review of systematic reviews of the health effects of performing arts participation;

2. A systematic review of observational studies investigating the impact of performing arts participation on mortality and non-communicable disease risk. *NB: initial searches revealed no prior systematic reviews addressing the effects of performing arts participation on mortality and/or non-communicable disease risk.*

3. A systematic review of studies of heart rate responses to performing arts participation.

Search terms and inclusion/exclusion criteria for each component are described below. All components involved searches of MEDLINE (*all fields*, *English & human subjects limiter*), EMBASE (*all fields*, *English & human subjects limiter*), SPORTDiscus (*all fields*, *English limiter*), and Web of Science (*Arts & Humanities citation index; all fields; English limiter)* from inception– 15 June 2020. Abstracts of all database search results were screened, followed by full text review of potentially relevant articles. Hand searches of the reference lists of included articles were also conducted to locate additional relevant articles. The review procedure was conducted by the first author in consultation with the authorship team.

Across all components, 'music participation' was defined as singing or playing a musical instrument. 'Dance participation' was broadly defined as an activity involving "moving one's body rhythmically. . .to music" [16], with an additional criterion that included articles must identify the investigated activity as 'dance.' Articles investigating music and dance participation conducted with an exertive aim (i.e. music or dance session(s) designed to elicit a target heart rate/rating of perceived exertion) were excluded to maintain review focus on performing arts vs. exercise participation. As noted in the introduction, performing arts participation is distinguished from exercise participation by its distinctly expressive, rather than exertive, focus; exertion is an intrinsic byproduct, not an objective.

## 2.3 Systematic review of systematic reviews of the health benefits of performing arts participation

**2.3.1 Database search terms.** Database searches were performed using the following search terms and a 'Reviews' limiter where available: ((music* OR danc* OR performing art* OR choir OR choral) AND (psycholog* OR biochem* OR immun* OR cognit* OR physical OR health)).

**2.3.2 Inclusion/Exclusion criteria.** Inclusion criteria were systematic reviews examining the health effects of active performing arts participation in healthy adults, adolescents or children. A 'systematic review' was defined based on Cochrane definitions [17] as a review conducted using explicit, reproducible methodology and aiming to comprehensively synthesize all available relevant evidence. Exclusion criteria were assessed at the primary study level within relevant reviews: 1) studies with qualitative data only; 2) studies in which performing arts participation was conducted with a target exercise intensity or heart rate–these studies were judged to evaluate exercise, rather than performing arts participation; 3) studies of long-term dance or music interventions in experienced dancers or musicians; 4) single-group observational studies characterizing experienced dancers or musicians.

Systematic reviews including a mixture of primary studies meeting and not meeting inclusion/exclusion criteria were included if:

- *Reviews in which study results were quantitatively synthesized (i.e. meta-analysis)*–The majority (>50%) of included studies examined active performing arts participation in healthy populations and met no exclusion criteria

    OR

- *Reviews in which results were narratively synthesized (i.e. descriptive synthesis of quantitative primary study results)*–The results of primary studies of active participation in healthy populations meeting no exclusion criteria could be extracted and re-synthesized for the purposes of this review.

**2.3.3 Data extraction.** Demographic and outcome data were extracted for all included reviews and their underlying primary studies meeting inclusion criteria and no exclusion criteria. For each outcome, the effect of performing arts participation was determined to be

'positive', 'negative', 'no effect', or 'unclear'. Designations of 'positive', 'negative' and 'no effect' were given in cases where clear links between changes in a parameter and a corresponding positive/negative health effect exist in healthy populations (e.g. shift from pro- to anti-inflammatory tone–positive effect; delayed pubertal onset–negative effect). An 'unclear' designation was given in cases where such links between changes in a parameter and health effects do not exist (e.g. acute increase in IL-6).

**2.3.4 GRADE quality of evidence appraisal.** The GRADE system was favored for this review because of its alignment with review aims and applicability to systematic reviews of systematic reviews [17]; GRADE is specifically "designed for reviews. . .that examine alternative management strategies" [18]. The GRADE system results in an appraisal of the quality of evidence supporting conclusions related to each outcome of interest—very low; low; moderate; high. Specific criteria and appraisal methodology are detailed in the S1 Appendix.

**2.3.5 Evidence synthesis.** To minimize the biasing effects of overlapping reviews, all outcomes from primary studies included in multiple reviews were only considered once. The lone exception to this was one outcome (flexibility–sit & reach) from one primary study of dance [19] which was included in multiple meta-analyses [20, 21] and thus considered twice. Re-calculation of meta-analyses to remove this duplication was not considered necessary due to consistent effects of dance on flexibility across 4 reviews considering 15 individual studies [20–23]. Common outcomes were first combined and assigned a grouped health effect and GRADE appraisal at the review level. Outcomes and GRADE appraisals were then combined across reviews and assigned a health effect and GRADE appraisal at the umbrella review level. Where appropriate, outcome results were stratified by music/dance participation, sex, age, GRADE appraisal, or instrument/style. Outcomes were categorized by domain–domains used to organize evidence of the health benefits of physical activity were used as an initial framework, with additional domains added as required [12]. Specific outcomes contained within each category are detailed in S1 Table in S1 Appendix.

## 2.4. Systematic review of observational studies investigating the impact of performing arts participation on mortality and non-communicable disease risk

Given an absence of known reviews of epidemiologic data regarding performing arts participation and the importance of these data in evaluating health effects, the authors made an a priori decision to conduct a separate systematic review.

**2.4.1 Search terms.** Databases were searched using the following terms: ((music* OR danc* OR performing art* OR choir OR choral) AND (mortality OR public health OR disease OR risk) AND epidemiology).

**2.4.2 Inclusion/exclusion criteria.** Inclusion criteria were observational studies investigating the relationship between performing arts participation and all-cause mortality or non-communicable disease risk and/or non-communicable disease risk factors (i.e. metabolic syndrome) in adults, adolescents or children. No exclusion criteria were defined.

**2.4.3 Evidence synthesis, GRADE appraisal and synthesis.** Conducted using an adaptation of the procedure detailed in sections 2.3.3–2.3.5, with included primary studies appraised individually and then synthesized at the level of this systematic review.

## 2.5 Systematic review of studies of heart rate responses to performing arts participation

**2.5.1 Search terms.** Database searches were performed using the following search terms: ((music* OR danc* OR performing art* OR choir OR choral) AND (load OR intensity OR heart rate)).

**2.5.2 Inclusion/Exclusion criteria.**   Inclusion criteria were studies reporting average/ mean heart rate data collected from at least 1 minute of a representative period of active music or dance participation in any setting. Studies reporting heart rate such that raw heart rate data (beats per minute) could not be extracted were excluded.

**2.5.3 Data extraction and appraisal.**   Demographic and raw heart rate data were extracted from all included studies. Raw heart rate data were calculated where necessary (i.e. from data expressed as % maximum heart rate). Rigorous application of inclusion/exclusion criteria was used in lieu of a formal assessment of evidence quality.

**2.5.4 Evidence synthesis.**   Raw heart rate data were converted to % heart rate maximum (%HR$_{max}$) using common estimation methods [24, 25]:

$$\%HR_{max} = ((\text{raw heart rate}/(208 - (0.7*\text{age}))*100).$$

%HR$_{max}$ values were then categorized by intensity according to American College of Sports Medicine definitions [26].

# 3. Results

## 3.1 Systematic review statistics (Fig 1)

This umbrella review includes 33 systematic reviews of the health effects of performing arts participation (15 dance; 18 music), encompassing 286 unique primary studies (128 dance; 158 music) and 149 outcomes across 18 health domains. Additionally, 9 observational studies investigating the impact of performing arts participation on mortality and non-communicable disease risk (3 dance, 5 music, 1 dance & music) were included, as well as 87 studies reporting heart rate responses during performing arts participation (71 dance, 16 music). Review articles and observational studies of mortality and non-communicable disease risk are directly referenced in this manuscript (Tables 1 and 2); the complete list of references, including studies investigating heart rate responses, is contained in the S1 Appendix.

## 3.2 General health effects of active performing arts participation

Positive effects of performing arts participation were reported in 17 of 18 investigated domains–only glucose/insulin outcomes were consistently reported to be unaffected by dance participation (no data related to music participation)(Table 1). Positive effects in 9 domains (*auditory; body composition; cognitive; immune function; mental health; physical fitness; physical function; self-reported health/wellbeing; social wellbeing*) were supported by moderate to high quality evidence; results in 4 of these 9 domains (*cognitive; mental health; physical fitness; self-reported health/wellbeing*) included a mixture of positive and neutral/no effects varying by specific outcome (Table 2). Positive effects of performing arts participation were found in 9 of 13 domains (7 of 13 supported by moderate-high quality evidence) associated with the mechanisms of physical activity benefits (Tables 1 and 2) [12, 28]. Raw data underpinning summary results and GRADE appraisals are detailed in the S1 Appendix.

Effects of performing arts participation were investigated in adult populations (*age 20–59*) across all domains backed by moderate-high quality evidence. Benefits of performing arts participation (*moderate-high quality evidence*) in children (*age 0–9*) and adolescents (*age 10–19*) were reported in auditory (music), body composition (dance), cognitive (music), and physical fitness (dance) domains; positive effects of dance on adolescent mental health were also reported. Benefits of performing arts participation were reported in older adults (*age ≥60; moderate-high quality evidence*) across cognitive (dance); immune function (music), mental

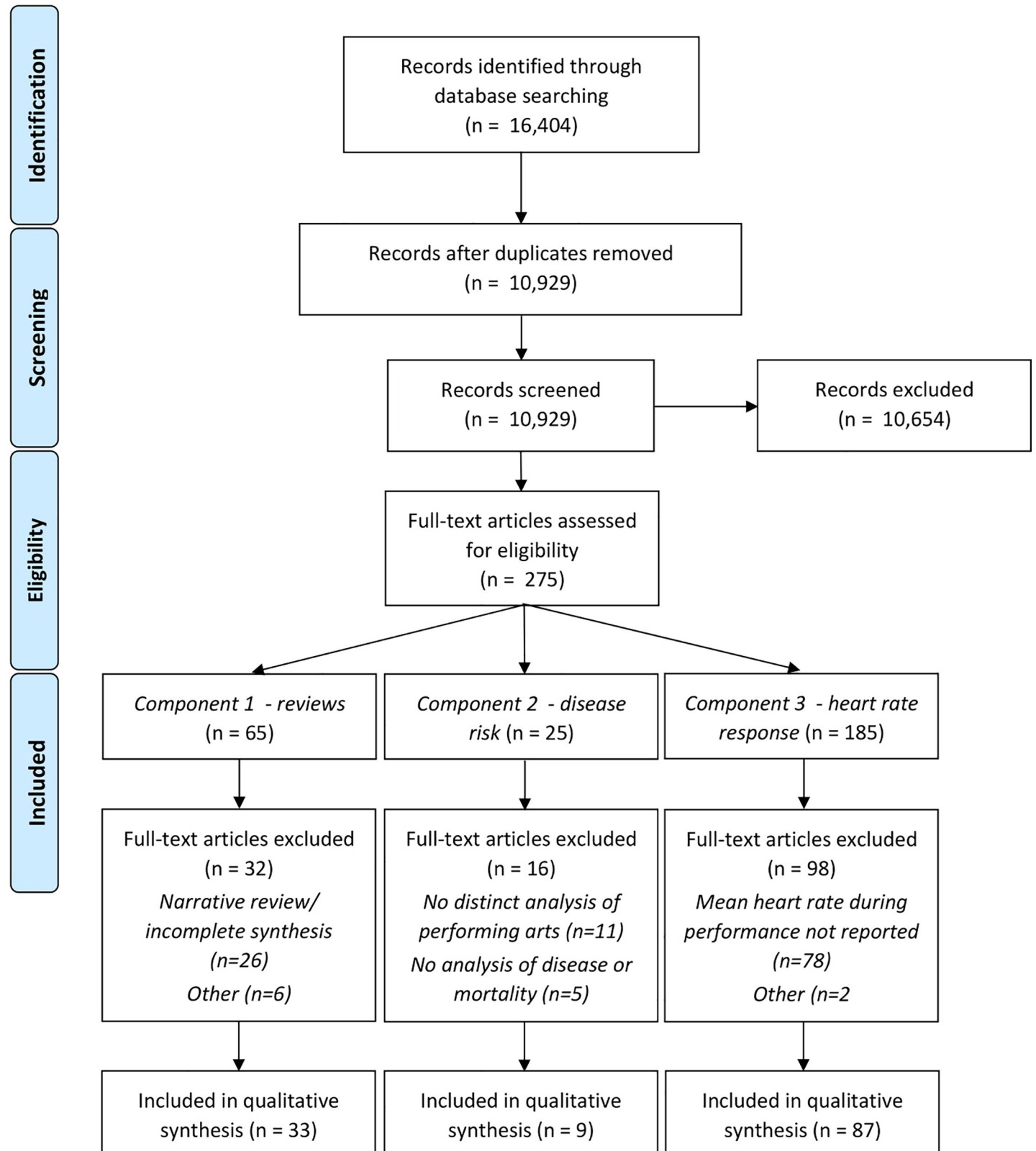

**Fig 1. PRISMA diagram [27] detailing umbrella review results.** Specific details regarding excluded reviews/studies are contained in the S1 Appendix.

**Table 1. Summary of effects of music and dance participation from included reviews and observational studies (non-communicable disease risk) on health parameters, grouped by domain.**

| Outcome Category | MUSIC PARTICIPATION | | DANCE PARTICIPATION | |
|---|---|---|---|---|
| | Effects | GRADE quality of evidence | Effect | GRADE quality of evidence |
| *Auditory [29, 30]* | Positive, No effect, Negative | Very Low, Low, Moderate | | |
| *Autonomic Tone [31]*[#] | Positive, No effect | Very Low | | |
| *Blood pressure [20, 32–35]*[#] | Positive, No effect | Very Low | Positive, No effect | Very Low |
| *Body Composition [20, 22, 23, 34–36]*[#] | | | Positive, No effect | Very Low, Moderate |
| *Bone Health [20, 22, 34, 37]* | | | Positive, No effect | Very Low, Low |
| *Cognitive [22, 29, 31, 32, 38–42]* | Positive, No effect | Very Low, Low, Moderate, High | Positive, No effect | Very Low, Low, Moderate, High |
| *Developmental (physical) [23, 34]* | | | Positive, No effect, Negative | Very Low, Low |
| *Educational [29, 41, 43–45]* | Positive, No effect, Negative | Very Low | | |
| *Stress Response/Endothelial function [30, 31, 33, 46]*[#] | Positive, No effect | Very Low, Low | | |
| *Glucose/Insulin [20, 22, 35]*[#] | | | No effect | High |
| *Immune function/Inflammation [30–32, 35, 46]*[#] | Positive, No effect | Very Low, Low, High | Unclear, No effect | Very Low, Low |
| *Lipid lipoprotein profile [20, 22, 35, 36]*[#] | | | Positive, No effect | Very Low |
| *Mental health [22, 30–34, 47–52]*[#] | Positive, No effect | Very Low, Low | Positive, No effect | Low, Moderate, High |
| *Non-communicable disease risk [53–61]* | Positive, No effect, Negative | Very Low, Low | Positive, No effect | Very Low, Low |
| *Physical fitness [20–23, 34–36, 38, 46, 62]*[#] | Positive, No effect | Very Low | Positive, No effect | Very Low, Low, Moderate, High |
| *Physical function [20–23, 30, 33, 34, 36, 38, 52, 62–65]*[#] | Positive, No effect | Very Low, Low | Positive, No effect | Very Low, Low, Moderate, High |
| *Self-reported health/wellbeing [21, 22, 30–34, 36, 38, 46, 49–52, 62, 66]* | Positive, No effect | Very Low, Low, High | Positive, No effect | Very Low, Low, Moderate, High |
| *Social functioning [30–33, 38, 47, 49–52]* | Positive, No effect | Very Low, Low, High | Positive | Moderate |

#—domain linked to mechanisms of the health benefits of physical activity (no performing arts data associated with 3 proposed domains/mechanisms–cardiac function; blood coagulation; coronary blood flow) [12, 28]. 'Positive' and 'no effect' results highlighted in green and black, respectively, are supported by moderate and/or high quality evidence.

health (dance), physical fitness (dance), physical function (dance), self-reported health/wellbeing (music & dance), and social functioning (music & dance) domains.

## 3.3 Modes and 'doses' of performing arts participation associated with reported health effects (Table 2; moderate–high quality evidence only)

The effects of dance participation were more broadly supported by higher quality evidence– 34 individual outcomes, with positive effects reported across 7 domains (*body composition; cognitive; mental health; physical fitness; physical function; self-reported health/wellbeing; social functioning*). The effects of music participation were supported by moderate to high quality evidence for 11 individual outcomes, with reported positive effects in 5 domains (*auditory; cognitive; immune function/inflammation; self-reported health/wellbeing; social functioning*). Modes of performing arts participation associated with the broadest positive health effects

**Table 2. Details of specific outcomes with moderate–high quality of evidence (GRADE).**

| Domain | Outcome | GRADE | Effect | # reviews | # studies/ outcomes | Sex | Age group | Music/ Dance? | Style/instrument | Participation Length |
|---|---|---|---|---|---|---|---|---|---|---|
| **Auditory** | Auditory processing | Moderate | Positive | 1 [29] | 13 | Mixed | Adults | Music | Instrumental | Sustained |
| | Pitch discrimination | Moderate | Positive | 1 [29] | 7 | Mixed | Adults | Music | Instrumental, unspecified | Sustained |
| | Speech in noise | Moderate | Positive | 1 [29] | 21 | Mixed | Children, Adolescents, Adults | Music | Instrumental, vocal, unspecified | Sustained |
| **Autonomic Tone**[#] | | | | | | | | | | |
| **Blood pressure**[#] | | | | | | | | | | |
| **Body Composition**[#] | Skinfold measurements | Moderate | Positive | 1 [20] | 3 | Female, Unspecified | Children, Adolescents, Adults | Dance | Aerobic dance | Sustained |
| | Total fat mass | Moderate | Positive | 1 [20] | 4 | Female | Children, Adolescents, Adults | Dance | Aerobic dance, Zumba | Sustained |
| **Bone Health** | | | | | | | | | | |
| **Cognitive** | IQ | Moderate | Positive | 2 [29, 41] | 5 | Mixed | Children, Adolescents, Adults | Music | Instrumental, Music education | Sustained |
| | Memory (long-/ short-term, working) | Moderate | Positive | 1 [42] | 42 | Mixed | Adults | Music | Instrumental | Sustained |
| | Spatial ability/ reasoning | High | Positive | 1 [40] | 23 | Mixed | Children, Adolescents | Music | Music education (general, Kodaly, Kindermusik, snare drum, piano, vocal) | Sustained |
| | Attention | Moderate | Positive | 1 [38] | 2 | Mixed | Older adults | Dance | Agilando, Multiple (line/jazz/ rock'n'roll/ square) | Sustained |
| | BDNF | Moderate | Positive | 1 [38] | 1 | Mixed | Older adults | Dance | Multiple (line/jazz/ rock'n'roll/ square) | Sustained |
| | Brain structure/ plasticity | Moderate | Positive | 1 [38] | 3 | Mixed | Adults, Older adults | Dance | Multiple (line/jazz/ rock'n'roll/ square), unspecified | Sustained |
| | Cognitive function/Global cognition | High | Positive | 2 [38, 39] | 10 | Mixed | Older adults | Dance | Agilando, Ballroom, Jazz, Latin, Tango, Square dance | Sustained |
| | Executive function | High | No effect | 3 [22, 38, 39] | 11 | Mixed | Older adults | Dance | Ballroom, Contemporary, Folk, Latin, Social, Tango, Waltz | Sustained |
| | Perceptual speed | High | No effect | 1 [38] | 2 | Mixed | Adults, Older adults | Dance | Social, unspecified | Sustained |
| | Vocabulary | High | No effect | 1 [38] | 1 | Mixed | Older adults | Dance | Social | Sustained |
| **Developmental** | | | | | | | | | | |
| **Educational** | | | | | | | | | | |
| **Stress Response / Endothelial function**[#] | | | | | | | | | | |
| **Glucose/Insulin**[#] | Glucose | High | No effect | 3 [20, 22, 35] | 6 | Mixed | Adults | Dance | Aerobic Dance, Ballroom, Bhangra, Zumba | Sustained |
| | Insulin | High | No effect | 2 [22, 35] | 2 | Female | Adults | Dance | Bhangra, Zumba | Sustained |

(*Continued*)

**Table 2.** (Continued)

| Domain | Outcome | GRADE | Effect | # reviews | # studies/ outcomes | Sex | Age group | Music/ Dance? | Style/instrument | Participation Length |
|---|---|---|---|---|---|---|---|---|---|---|
| *Immune function /Inflammation*[#] | Immunological / inflammatory profile | High | Positive | 2 [31, 32] | 2 | Mixed | Adults | Music | Drums | Acute |
| | Immunoglobulin A | High | Positive | 3 [30, 31, 46] | 4 | Mixed | Adults, Older adults | Music | Singing, Drums | Acute |
| *Lipid lipoprotein profile*[#] | | | | | | | | | | |
| *Mental health*[#] | Depression | High | No effect | 1 [22] | 1 | Mixed | Older adults | Dance | Turkish folk dance, Jazz, Social | Sustained |
| | Mood | Moderate | Positive | 1 [34] | 1 | Mixed | Adults | Dance | Hip hop | Acute |
| | Self-perception | Moderate | Positive | 1 [34] | 2 | Female | Adolescents | Dance | Aerobic dance | Sustained |
| *Non-communicable disease risk* | | | | | | | | | | |
| *Physical fitness*[#] | Abdominal strength/ endurance (sit ups) | Moderate | Positive | 2 [20, 34] | 5 | Female, Unspecified | Children, Adolescents, Adults | Dance | Aerobic dance | Sustained |
| | Cardiovascular capacity (VO$_2$ max) | High | Positive | 3 [20, 34, 35] | 12 | Mixed | Adolescents, Adults, Older Adults | Dance | Aerobic dance, Balinese, Dance Team, Greek folk/traditional dance, Waltz, Zumba | Sustained |
| | Endurance (6-minute walk test) | High | Positive | 3 [20, 21, 35] | 6 | Mixed | Adults, Older adults | Dance | Aerobic Dance, Ballroom, Thai, Turkish folk, Zumba | Sustained |
| | Power (muscular/ aerobic) | High | No effect | 2 [22, 23] | 2 | Mixed | Adults, Older adults | Dance | Ballet, Salsa | Sustained |
| | Peak ventilation | High | Positive | 1 [20] | 4 | Mixed | Adults, Older adults | Dance | Aerobic dance, Greek folk/traditional dance, Zumba | Sustained |
| | Respiratory exchange ratio | Moderate | No effect | 1 [20] | 2 | Mixed | Adults | Dance | Aerobic dance, Zumba | Sustained |
| | Strength | Moderate | Positive | 3 [20, 22, 34] | 8 | Mixed | Children, Adolescents, Adults, Older adults | Dance | Aerobic Dance, Dance Team, Social | Sustained |
| *Physical function*[#] | Balance | High | Positive | 6 [21, 22, 38, 63–65] | 47 | Mixed | Adolescents, Adults, Older adults | Dance | Aerobic Dance, Agilando, Ballet, Ballroom, Caribbean, Contemporary, Greek traditional, Latin, Lebed Method, Line dance, Modern, Multiple (line/jazz/rock'n'roll/square), Opera, Salsa, Thai, Turkish folk, Zumba | Sustained |
| | Flexibility/range of motion | High | Positive | 4 [20–23] | 19 | Mixed | Children, Adolescents, Adults, Older Adults | Dance | Aerobic dance, Ballroom, Ballet, Folk/traditional dance, Social, Thai, Zumba | Sustained |
| | Mobility (timed up & go; sit to stand) | Moderate | Positive | 2 [21, 22] | 12 | Mixed | Older adults | Dance | Aerobic dance, Argentine Tango, Ballroom, Folk, Lebed method, Turkish, Thai | Sustained |
| | Proprioception | High | Positive | 1 [65] | 1 | Mixed | Older adults | Dance | Creative Dance | Sustained |

*(Continued)*

**Table 2.** (Continued)

| Domain | Outcome | GRADE | Effect | # reviews | # studies/ outcomes | Sex | Age group | Music/ Dance? | Style/instrument | Participation Length |
|---|---|---|---|---|---|---|---|---|---|---|
| Self-reported health/wellbeing | Fatigue | High | Positive | 2 [20, 33] | 2 | Mixed | Adults | Music | Drums | Acute |
| | Quality of life | High | Positive | 2 [33, 49] | 2 | Mixed | Adults, Older Adults | Music | Singing | Sustained |
| | Alcohol Consumption | Moderate | No effect | 1 [22] | 1 | Mixed | Older adults | Dance | Caribbean | Sustained |
| | Balance confidence | High | Positive | 1 [22] | 1 | Mixed | Older adults | Dance | Argentine tango | Sustained |
| | Functional autonomy | High | Positive | 1 [22] | 1 | Unspecified | Older adults | Dance | Ballroom | Sustained |
| | Life satisfaction | High | Positive | 1 [22] | 1 | Mixed | Older adults | Dance | Creative Dance | Sustained |
| | Sexual activity | Moderate | Positive | 1 [22] | 1 | Mixed | Older adults | Dance | Caribbean | Sustained |
| | Sleep quality | Moderate | Positive | 1 [22] | 1 | Mixed | Older adults | Dance | Caribbean | Sustained |
| | Smoking | Moderate | No effect | 1 [22] | 1 | Mixed | Older adults | Dance | Caribbean | Sustained |
| | Stress | Moderate | Positive | 1 [38] | 1 | Mixed | Older adults | Dance | Social | Sustained |
| Social functioning | Anger | High | Positive | 2 [31, 33] | 2 | Mixed | Adults, Older adults | Music | Drums/Percussion & keyboard education | Acute/ Sustained |
| | Social Support (perceived) | Moderate | Positive | 1 [38] | 1 | Mixed | Older adults | Dance | Social | Sustained |

Age group classifications based on United Nations/World Health Organization definitions: 0–9 years–children; 10–19 years–adolescents; 20–59 –adults; 60+–older adults. 'Acute' participation refers to a single session (up to 2.5 hours) of performing arts participation; 'sustained' participation refers to 4+ weeks of at least weekly performing arts participation.

#—domain linked to mechanisms of the health benefits of physical activity (no performing arts data associated with 3 proposed domains/mechanisms–cardiac function; blood coagulation; coronary blood flow). [12, 28].

were: aerobic dance (4 *domains*); ballroom dance (4 *domains*); social dance (4 *domains*); drumming (3 *domains*); and Zumba dance (3 *domains*).

Acute doses (*single session lasting 30–60 minutes*) were sparsely associated with positive effects—hip-hop dance benefited mental health (mood) and music participation (drumming; singing) was associated with positive changes in immune function/inflammation, self-reported health/wellbeing (fatigue), and social functioning (anger). All other results were based on studies of sustained performing arts participation. Significant heterogeneity in frequency and timing of sustained participation was found. Positive health effects were associated with sustained performing arts participation lasting at least 4 weeks, with a minimum of 60 minutes of weekly participation and at least one weekly session. Each individual session in intervention studies lasted 21–120 minutes; the length of individual sessions in cross-sectional studies of performing arts participants vs. non-participants was generally not reported.

### 3.4 Physical demands of performing arts participation

Heart rate responses to performing arts participation widely varied by style and/or performance setting, with studies of both music and dance participation reporting heart rates classified as very light, light, moderate, and vigorous intensity physical activity (Tables 3 and 4). Heart rate also varied substantially within the same mode of music/dance participation, with 16 modes (12 music; 4 dance) associated with heart rate responses at two intensity levels, 3 modes (1 music–trumpet; 2 dance–ballet, modern) associated with heart rate responses at three intensity levels, and active

**Table 3. Summary of heart rate responses to active music participation from included studies.**

| | Instrument/style | Participation Setting |
|---|---|---|
| *Very Light (<57% max)* | Classical Indian Music | Performance |
| | Contemporary band* | Rehearsal |
| | Drum corps* | Rehearsal |
| | Flute/Singing* | Rehearsal |
| | Marching band* | Rehearsal |
| | Piano* | Rehearsal |
| | Strings* | Rehearsal, Practice |
| | Trumpet** | Practice |
| | Varied instruments in orchestra* | Rehearsal, Performance |
| | Winds* | Rehearsal |
| *Light (57–63% max)* | Clarinet* | Performance |
| | Contemporary band* | Performance |
| | Drum corps* | Rehearsal |
| | Percussion (classical) | Performance |
| | Singing (operetta)* | Performance |
| | Strings* | Performance |
| | Trumpet** | Laboratory |
| | Winds* | Performance |
| *Moderate (64–76% max)* | Bagpipes* | Laboratory |
| | Clarinet* | Performance |
| | Conductor (opera) | Performance |
| | Drum set* | Laboratory |
| | Flute/singing* | Performance |
| | Marching band* | Performance |
| | Singing (Opera) | Performance |
| | Piano* | Performance |
| | Trumpet** | Laboratory |
| | Varied instruments in orchestra* | Performance ('public session') |
| *Vigorous (≥77% max)* | Bagpipes* | Laboratory |
| | Drum set* | Performance |
| | Musical theater (singing + dance) | Laboratory |
| | Singing (operetta)* | Performance |

*—instruments / styles with reported heart rate responses at 2 intensity levels.

**—instruments/styles with reported heart rate responses at 3 intensity levels. See S1 Appendix for source data and citations.

video game dancing associated with heart rate responses at all four intensity levels. Raw heart rate data underpinning summary results are detailed in the S1 Appendix.

## 4. Discussion

This umbrella review presents an expansive and detailed synthesis and appraisal of evidence demonstrating that performing arts participation is, broadly, health promoting activity, with positive effects across 17 health domains. Moderate-high quality evidence supported positive effects across 9 of these domains, including 7 of 13 domains associated with the health benefits of physical activity. Positive effects were reported in adult populations across all 9 domains, with beneficial effects in children, adolescents, and older adults reported across 4, 5, and 7

**Table 4. Summary of heart rate responses to active dance participation from included studies.**

| | Dance style | Participation setting |
|---|---|---|
| **Very Light (<57% max)** | Active Video Game Dance*** | Laboratory |
| | Modern** | Rehearsal/Class |
| **Light (57–63% max)** | Active Video Game Dance*** | Laboratory |
| | Ballet** | Class |
| | Fox trot | Class |
| | Merengue | Class |
| | Mixed ('Dancing Classrooms') | Class |
| | Modern** | Class |
| | Rhumba | Class |
| | Salsa* | Class |
| | Tango | Class |
| | Waltz | Class |
| **Moderate (64–76% max)** | Active Video Game Dance*** | Laboratory |
| | Aerobic Dance* | Laboratory, Class |
| | Ballet** | Class |
| | Dance Fitness Class | Laboratory |
| | Disco | Party |
| | Fijian | Laboratory |
| | 'Fun Dance' | Class |
| | Hawaiian Hula* | Laboratory |
| | Latin | Laboratory |
| | Line dancing | Class |
| | Maori haka | Laboratory |
| | Maori poi balls | Laboratory |
| | Mixture (anti-aging focus) | Class |
| | Modern** | Class, Dress Rehearsal |
| | Pole Dancing | Class |
| | Salsa* | Class, nightclub |
| | Samoan sasa | Laboratory |
| | Swing | Class |
| | Tongan | Laboratory |
| | Zumba* | Class, Home, Laboratory |
| **Vigorous (≥77% max)** | Active Video Game Dance*** | Laboratory |
| | Aerobic Dance* | Laboratory, Class |
| | Ballet** | Class, Rehearsal, Laboratory, Performance |
| | Ballroom | Laboratory |
| | Highland Dance | Rehearsal, Performance |
| | Hip-hop | Laboratory |
| | Hawaiian Hula* | Laboratory |
| | Musical Theater (dance only) | Laboratory |
| | Polish folk dancing | Laboratory |
| | Samoan slap | Laboratory |
| | Sardinian folk dance (ballu sardu) | Laboratory |
| | Sports Dancing | Laboratory |
| | Swedish folk dance (hambo) | Laboratory |
| | Tahitian | Laboratory |
| | Tap dance | Laboratory |
| | Tinikling (traditional Filipino dance) | Laboratory |
| | Zumba* | Class |

*—dance styles with reported heart rate responses at 2 intensity levels

**—dance styles with reported heart rate responses at 3 intensity levels

***—dance styles with reported heart rate responses at all 4 intensity levels. See S1 Appendix for source data and citations.

domains, respectively. This review also provides preliminary insights into the modes and doses of performing arts participation underpinning observed benefits. Further, heart rate data from 87 additional studies indicate that both music and dance participation intrinsically elicit mean heart rate values corresponding to a range of intensities, including moderate and vigorous.

This review also reveals that the evidence regarding the health impacts of performing arts participation is still in its infancy. Accordingly, reported health benefits and preliminary insights regarding effective performing arts modes and doses must be considered within this context. Moderate-high quality results provide valuable guidance but should not be interpreted as supporting the totality of health benefits or superiority of modes or doses of performing arts participation. Key results of this review are due to greater amounts of high-quality studies of specific modes and doses in particular domains; the overall quality of included evidence is generally low (*26% (45/173) of outcomes backed by moderate-high quality evidence*) due to a predominance of non-randomized and observational vs. randomized controlled trial study designs. Control and comparison groups varied widely across all study types, including no-intervention/waitlist control groups and exercise (various types), cognitive and/or language training and other art participation (e.g. visual art, drama) comparison groups.

All included studies were conducted without explicit intensity aims, yet 2 of 4 modes of dance participation associated with the broadest health benefits come from exercise, not artistic, traditions: aerobic dance [67] and Zumba [68]. Heart rate data (Table 4) unsurprisingly confirm that both modes are associated with moderate to vigorous intensity physical demands as per global physical activity recommendations [69]. However, two traditionally expressive modes of dance–ballroom and social–were found to have similarly broad benefits, including in physical fitness and function domains. Heart rate data for ballroom and social dancing were sparsely available, precluding discussion of the potential impact of intrinsic physical intensity. Nonetheless, these results suggest that expressive dance participation is similarly health promoting to modes created from an exercise viewpoint. Further obscuring the relationships between physical intensity and observed benefits, drumming was the most broadly health promoting mode of music participation and associated, across various settings, with very light, light, moderate, and vigorous intensity heart rate responses. Additional research is needed to establish the relationships between intrinsic physical intensity and health impacts during performing arts participation.

Both acute and sustained performing arts participation were associated with health benefits, although the bulk of evidence relates to sustained participation. Acute benefits of singing and drumming on inflammation and immune parameters are particularly intriguing; similar short-term effects have been associated with physical activity and linked, with sustained participation, to long-term preventive benefits [70]. Epidemiologic studies suggest similar links between sustained performing arts participation and a reduced risk of non-communicable diseases and early mortality. However, the quality of these epidemiologic studies is presently low and specifically limited by the use of a range of bespoke survey instruments with unclear psychometric value to quantify the frequency, timing/duration and type of performing arts participation. Future studies using validated instruments for quantifying performing arts participation are needed.

The majority of health benefits backed by moderate-high quality evidence were associated with sustained performing arts participation lasting at least four weeks. Although substantial heterogeneity in results limits conclusions regarding the impact of specific doses of the performing arts, all reported benefits were associated with at least weekly participation. Some benefits were seen with as little as 60 minutes of weekly participation, demonstrating that, like physical activity, significant health benefits can be achieved with modest effort and time

commitment [14]. Physical activity evidence indicates that greater levels of weekly participation are associated with greater health benefits–'*some is better than none, more is better than less' [14]*. Substantial further research is required to determine the impact of the frequency and duration of performing arts participation on health benefits, as well as the potential additional impact of the setting (e.g. laboratory, classroom, live performance) of performing arts participation on observed benefits.

In sum, this review presents promising evidence regarding the health benefits of performing arts participation, but is limited by a young and disparate evidence base, as well as additional factors discussed below. Excepting studies of non-communicable disease risk, this umbrella review was limited to English language studies included in systematic reviews of the health effects of performing arts participation. It is thus probable that some primary studies were not considered; their exclusion could impact individual outcome results given the aforementioned infancy of the evidence base. However, it is less likely that individual primary studies would significantly impact the general conclusions of this review, which are based on aggregated moderate-high quality evidence grouped by domain of health impact.

This review is also potentially limited by the conduct of literature searches, data extraction, and evidence appraisal by the first author alone, in consultation with the authorship team, due to resource constraints. Single author search, extraction and appraisal has been demonstrated to increase the incidence of errors [71], yet these errors have been found to have a minimal impact on review results and conclusions [72]. To best meet study aims, the authors thus favored a broad, single author search, extraction and appraisal over a more constrained review conducted by multiple authors in duplicate. Additionally, the inclusion of a comprehensive and transparent S1 Appendix detailing all review data and subjective decision-making (i.e. article inclusions, GRADE appraisals) clarifies the basis for specific conclusions and serves as a foundation for discussion and future research.

Finally, while studies of participation-related performing arts injuries were beyond the scope of this review, it should be noted that, similar to exercise participation [73], the health impact of performing arts activities is not exclusively positive. Participation in performing arts does carry an injury risk, for example caused by overpractice [10]. These risks are considerably counterbalanced by the broad benefits of performing arts participation demonstrated in this review. However, on an individual level, participation risks must always be managed and weighed against potential benefits.

## 4.1 Conclusions

Performing arts participation is, broadly, a health promoting activity, with beneficial effects reported across healthy (non-clinical) children, adolescents, adults, and older adults in 17 domains (9 supported by moderate-high quality evidence). Positive health effects were associated with as little as 30 (*acute participation*) or 60 (*sustained weekly participation*) minutes of performing arts participation, with drumming and both expressive (*ballroom*, *social*) and exercise-based (*aerobic dance*, *Zumba*) modes of dance linked to the broadest health benefits. However, the evidence base is still very much in its infancy. Further research is necessary to optimize modes and doses of performing arts participation towards specific health effects, as well as clarify relationships between intrinsic physical intensity and observed benefits. The broad yet rigorous approach of this umbrella review provides a valuable knowledge foundation for such future research.

## Supporting information

**S1 Appendix. The S1 Appendix contains raw data underpinning GRADE appraisals and summary results of reviews, epidemiologic studies, and heart rate data.** Additionally, details of excluded articles are included.
(PDF)

## Author Contributions

**Conceptualization:** J. Matt McCrary.

**Data curation:** J. Matt McCrary, Eckart Altenmüller.

**Formal analysis:** J. Matt McCrary, Emma Redding.

**Funding acquisition:** J. Matt McCrary, Eckart Altenmüller.

**Investigation:** J. Matt McCrary.

**Methodology:** J. Matt McCrary, Eckart Altenmüller.

**Project administration:** J. Matt McCrary.

**Supervision:** Eckart Altenmüller.

**Writing – original draft:** J. Matt McCrary.

**Writing – review & editing:** J. Matt McCrary, Emma Redding, Eckart Altenmüller.

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
