## [Decision Letter · Decision Letter 0]

1 Apr 2021

PONE-D-20-39389

Performing arts as a health resource? An umbrella review of the health impacts of music and dance participation

PLOS ONE

Dear Dr. McCrary,

Thank you for submitting your manuscript to PLOS ONE. After careful consideration, we feel that it has merit but does not fully meet PLOS ONE’s publication criteria as it currently stands. Therefore, we invite you to submit a revised version of the manuscript that addresses the points raised during the review process.

As you can see, both reviewers are generally supportive of this work, however several concerns they raise must be concerned. I felt like this was somewhere between a major and minor revision, but in order to give you more time to complete the revisions, I've decided to issue this as a major revision (but in reality, I am hoping it won't take that long to address the reviewers' comments and resubmit).

We look forward to receiving your revised manuscript.

Kind regards,

Emily S. Cross

Academic Editor

PLOS ONE

2. In line with PLOS' guidelines on systematic reviews (https://journals.plos.org/plosone/s/submission-guidelines#loc-systematic-reviews-and-meta-analyses), please update your PRISMA flowchart to provide detailed reasons for the exclusion of manuscripts at each stage of analysis.

*As per PLOS' guidelines regarding the reporting of Materials and Methods (https://journals.plos.org/plosone/s/submission-guidelines#loc-materials-and-methods), authors should ensure that "the Materials and Methods section should provide enough detail to allow suitably skilled investigators to fully replicate your study." Specifically, please consider moving some methodological details from the Supplemental information to the main body of the manuscript, so that your Methods section includes sufficient information to be understood independently of any other methods or measures described elsewhere in your submission.

Reviewers' comments:

Reviewer's Responses to Questions

**Comments to the Author**

1. Is the manuscript technically sound, and do the data support the conclusions?

Reviewer #1: Yes

Reviewer #2: Yes

2. Has the statistical analysis been performed appropriately and rigorously? 

Reviewer #1: Yes

Reviewer #2: N/A

3. Have the authors made all data underlying the findings in their manuscript fully available?

Reviewer #1: Yes

Reviewer #2: Yes

4. Is the manuscript presented in an intelligible fashion and written in standard English?

Reviewer #1: Yes

Reviewer #2: Yes

5. Review Comments to the Author

Reviewer #1: The current paper provides an indepth review of the health benefits of participation in and engagement with the performing arts. The results show that both music, especially drumming, and expression and exercise-based modes of dance promote health, but specific conclusions are as yet limited. I enjoyed reading the paper and the research questions are important and timely, with the potential to contribute to the field. However, I believe that a few things can be further clarified, and a few concerns need to be addressed before the paper goes forward for publication. I outline both minor and major comments below. For convenience, I outline these chronologically.

1. Page 3, line 63. Please clarify in the introduction what you mean by the ‘mode’ of art.

2. I think it would be helpful in the introduction to clarify front and center that there are a number of performing arts and the authors chose to focus on music and dance, and provide more justification for this choice.

3. Additionally, it would be beneficial for the reader if the authors provide their apriori hypotheses in the introduction itself. What were your predictions and why?

4. Page 4, line 97: “Following informal literature searches, the authors made an a priori decision that an integrated, three component umbrella review would most effectively address study aims” – can you clarify more precisely in the introduction what exactly your study aims were and why you focused on only these three components. Aren’t there other components that can also answer your study aims? I think a more comprehensive justification would be helpful.

5. Page 5, line 117: Please clarify what you mean by an ‘exertive aim.’ Was the exclusion criteria such that studies were excluded if target heart rate or perceived exertion were dependent variables? Can you please justify more why you chose to exclude these studies but include a separate systematic review on heart rate responses? To clarify, I understand why the authors did this but I think the readers would benefit from a clearer justification.

6. Page 6, line 143: it is unclear from the phrasing whether only those systematic reviews where meta-analyses were conducted were included in the current review or whether all systematic reviews were included?

7. Page 6, line 147: please clarify what you mean by narrative synthesis.

8. Page 7, line 54: “For each outcome, the effect of performing arts participation was determined to be ‘positive’, ‘negative’, ‘no effect’, or ‘unclear’” Please clarify in more detail how each of these outcomes were determined.

9. Page 7, line 167: It is not clear why one outcome from one study which was included in multiple meta-analyses was considered twice. Please clarify this.

10. Page 7, line 172: “similar domains associated with the health benefits of physical activity” – why were similar domains chosen? Was the goal of the study to find similar outcomes to any other physical activity including dance and music, or was it to find health outcomes that were specific to dance and music above and beyond health outcomes of just any physical activity? I think this is a concern throughout the paper where the aim is slightly unclear in terms of identifying outcomes that are specific to expressive activity and that are outcomes of any exercise.

11. Page 10, Figure 1: Figure 1 is excellent and a graphical representation of your umbrella review. It would be more helpful to the reader either in text or in the caption to add the specific reason why studies were excluded (Eg. Xx studies excluded because no heart rate data, xx studies excluded because included only in qualitative data etc).

12. Page 15, table 2: this is an excellent, extensive list of studies! Were the studies also grouped by style/instrument of dance or music in order to find whether combined together these studies yielded consistent (intensity) of health outcomes? Were the studies also grouped by participation setting (lab/class/performance etc) to see if there were participation setting specific effects?

13. Page 16 (for some reason line numbers disappear after table 2 � ): Did you have age-specific predictions?

14. Page 22, first paragraph: why is the quality of existing studies low? Is this because of low sample sizes? Or missing control groups? More broadly, I think the discussion needs to address the issue of control groups. What control groups were used in primary studies to dissociate the effects of music and/or dance from the effects of other forms of activity?

15. Page 22, first paragraph: “validated, sensitive instruments for quantifying performing arts participation” – please clarify what you mean by validated and sensitive instruments, and highlight directs for future research more broadly in the discussion that follow on from your findings.

Reviewer #2: Summary

This is a well-written and clear review of an area lacking important research. This review thoroughly collates the present evidence on the health impacts of music and dance participation and provides a solid foundation for future research.

General points

In particular it is commendable that music and dance are both covered in this review. I think it would be nice to acknowledge in the introduction their common evolutionary origins. The review supports the idea that group dancing and musicking fulfil similar psychological and societal functions https://www.pnas.org/content/112/29/8987.short

https://royalsocietypublishing.org/doi/10.1098/rsbl.2015.0767

https://journals.plos.org/plosone/article?id=10.1371/journal.pone.0180101

https://www.ncbi.nlm.nih.gov/pmc/articles/PMC6092630/

Specific points

1) Page, 3, first paragraph, a recent systematic review on dance interventions for mental health was published by Millman et al., 2020 here https://onlinelibrary.wiley.com/doi/full/10.1002/cpp.2490

2) Page 20, line 1/2 – “This umbrella review…” and “…evidence demonstrating that performing arts participation is, broadly, a health promoting activity…”

3) Page 21 line 1 – “…greater amounts of high quality studies of specific…”

4) Page 23, line 18 – “Performing arts participation is a broadly healthy promoting activity, with…” or could do the same as what I suggested for Page 20 “… is, broadly, a health promoting…”

6. PLOS authors have the option to publish the peer review history of their article (what does this mean?). If published, this will include your full peer review and any attached files.

Reviewer #1: No

Reviewer #2: No

---

## [Author Response · Author response to Decision Letter 0]

15 Apr 2021

Thank you for the opportunity to submit revisions of our manuscript for further consideration for publication in PLoS One. We thank the reviewers for their thorough feedback, which has helped us further strengthen our manuscript. Please find a point-by-point response to the reviewers’ comments below:

Reviewer #1: The current paper provides an indepth review of the health benefits of participation in and engagement with the performing arts. The results show that both music, especially drumming, and expression and exercise-based modes of dance promote health, but specific conclusions are as yet limited. I enjoyed reading the paper and the research questions are important and timely, with the potential to contribute to the field. However, I believe that a few things can be further clarified, and a few concerns need to be addressed before the paper goes forward for publication. I outline both minor and major comments below. For convenience, I outline these chronologically.

1. Page 3, line 63. Please clarify in the introduction what you mean by the ‘mode’ of art.

Author Response: The ‘mode’ of art has been clarified on p.3, lines 63-64 of the revised manuscript. The revised sentence now reads: ‘The specifics of the most effective arts interventions – namely the mode (specific ‘type’ of art – e.g. ballroom dance, singing) and ‘dose’ (frequency and timing/duration) – for various clinical and public health scenarios are still unclear.’

2. I think it would be helpful in the introduction to clarify front and center that there are a number of performing arts and the authors chose to focus on music and dance, and provide more justification for this choice.

Author Response: The revised manuscript now includes a more detailed justification for the focus of the review on performing arts participation, also incorporating suggestions from Reviewer #2. Page 3, lines 67-73 of the introduction now read:

Participation in the performing arts is the most popular form of arts participation, with up to 40% of EU and US adults participating annually in performing arts activities.[5, 6] Within the performing arts, music and dance participation are the two most popular modes of engagement, both involving engagement with music, and proposed to have common evolutionary origins.[7-9] The health effects of music and dance participation were thus considered likely to be both related and broadly studied and are the focus of this review. ‘Performing arts participation’ will be used to refer, jointly, to music and dance participation from this point forward.

3. Additionally, it would be beneficial for the reader if the authors provide their apriori hypotheses in the introduction itself. What were your predictions and why?

Author Response: A priori hypotheses for the review have been added to the revised introduction on p. 4, lines 92-94: ‘Performing arts participation is hypothesized to have similar health effects as physical activity due to its intrinsic physical exertion, as well as additional effects related to creative expression and engagement with music.’

4. Page 4, line 97: “Following informal literature searches, the authors made an a priori decision that an integrated, three component umbrella review would most effectively address study aims” – can you clarify more precisely in the introduction what exactly your study aims were and why you focused on only these three components. Aren’t there other components that can also answer your study aims? I think a more comprehensive justification would be helpful.

Author Response: The study aims have been clarified in the revised introduction and further justification for the three selected review components has been added to the revised methods. Clarified aims in the introduction now read (p. 4, lines 86-97):

Evidence regarding the full breadth of health impacts of performing arts participation, as well as the modes and doses underpinning these effects, has yet to be compiled, critically appraised and analyzed using a common framework. This umbrella review aims to address this knowledge gap by systematically reviewing and appraising evidence regarding the health effects of performing arts participation, including its impacts on both broad mortality and disease risk and more discrete health-related outcomes, in healthy (non-clinical) adults, adolescents and children. Performing arts participation is hypothesized to have similar health effects as physical activity due to its intrinsic physical exertion, as well as additional effects related to creative expression and engagement with music. Accordingly, a secondary aim of this review is to compile data regarding the intrinsic physical intensity of varying modes of performing arts participation to inform further hypotheses related to relationships between physical intensity and observed effects.

Section 2.2 of the revised methods now further clarifies the choice of the three umbrella review components selected to meet umbrella review aims (p. 5, lines 104-112):

Following informal literature searches, the authors made an a priori decision that an integrated, three component umbrella review would most effectively address study aims: 

1) A systematic review of systematic reviews of the health effects of performing arts participation; 

2) A systematic review of observational studies investigating the impact of performing arts participation on mortality and non-communicable disease risk. NB: initial searches revealed no prior systematic reviews addressing the effects of performing arts participation on mortality and/or non-communicable disease risk. 

3) A systematic review of studies of heart rate responses to performing arts participation.

5. Page 5, line 117: Please clarify what you mean by an ‘exertive aim.’ Was the exclusion criteria such that studies were excluded if target heart rate or perceived exertion were dependent variables? Can you please justify more why you chose to exclude these studies but include a separate systematic review on heart rate responses? To clarify, I understand why the authors did this but I think the readers would benefit from a clearer justification.

Author Response: Exclusion criteria related to studies conducted with an exertive vs. expressive aim have been clarified in the revised methods (p. 5-6, lines 125-130):

Articles investigating music and dance participation conducted with an exertive aim (i.e. music or dance session(s) designed to elicit a target heart rate/rating of perceived exertion) were excluded to maintain review focus on performing arts vs. exercise participation. As noted in the introduction, performing arts participation is distinguished from exercise participation by its distinctly expressive, rather than exertive, focus; exertion is an intrinsic byproduct, not an objective. 

6. Page 6, line 143: it is unclear from the phrasing whether only those systematic reviews where meta-analyses were conducted were included in the current review or whether all systematic reviews were included?

Author Response: The phrasing in this section has been clarified in the revised methods (p. 7; lines 151-160):

Systematic reviews including a mixture of primary studies meeting and not meeting inclusion/exclusion criteria were included if:

‒ Reviews in which study results were quantitatively synthesized (i.e. meta-analysis) – The majority (>50%) of included studies examined active performing arts participation in healthy populations and met no exclusion criteria; 

OR

‒ Reviews in which results were narratively synthesized (i.e. descriptive synthesis of quantitative primary study results) – The results of primary studies of active participation in healthy populations meeting no exclusion criteria could be extracted and re-synthesized for the purposes of this review.

7. Page 6, line 147: please clarify what you mean by narrative synthesis.

Author Response: The term narrative synthesis has been clarified on p. 7, lines 157-158 of the revised methods: ‘Reviews in which results were narratively synthesized (i.e. descriptive synthesis of quantitative primary study results)…’

8. Page 7, line 54: “For each outcome, the effect of performing arts participation was determined to be ‘positive’, ‘negative’, ‘no effect’, or ‘unclear’” Please clarify in more detail how each of these outcomes were determined.

Author Response: Further details regarding the determination of positive/negative/no effect/unclear designations of results has been added to the revised methods (p. 7, lines 165-171):

For each outcome, the effect of performing arts participation was determined to be ‘positive’, ‘negative’, ‘no effect’, or ‘unclear’. Designations of ‘positive’, ‘negative’ and ‘no effect’ were given in cases where clear links between changes in a parameter and a corresponding positive/negative health effect exist in healthy populations (e.g. shift from pro- to anti-inflammatory tone – positive effect; delayed pubertal onset – negative effect). An ‘unclear’ designation was given in cases where such links between changes in a parameter and health effects do not exist (e.g. acute increase in IL-6).

9. Page 7, line 167: It is not clear why one outcome from one study which was included in multiple meta-analyses was considered twice. Please clarify this.

Author Response: Treatment of the one outcome included in multiple meta-analyses has been clarified in the revised methods (p. 8, lines 182-187):

To minimize the biasing effects of overlapping reviews, all outcomes from primary studies included in multiple reviews were only considered once. The lone exception to this was one outcome (flexibility – sit & reach) from one primary study of dance [18] which was included in multiple meta-analyses [19, 20] and thus considered twice. Re-calculation of meta-analyses to remove this duplication was not considered necessary due to consistent effects of dance on flexibility across 4 reviews considering 15 individual studies.[19-22]

10. Page 7, line 172: “similar domains associated with the health benefits of physical activity” – why were similar domains chosen? Was the goal of the study to find similar outcomes to any other physical activity including dance and music, or was it to find health outcomes that were specific to dance and music above and beyond health outcomes of just any physical activity? I think this is a concern throughout the paper where the aim is slightly unclear in terms of identifying outcomes that are specific to expressive activity and that are outcomes of any exercise.

Author Response: This sentence has been revised for clarity (p.8, lines 192-194): ‘Outcomes were categorized by domain – domains used to organize evidence of the health benefits of physical activity were used as an initial framework, with additional domains added as required.[12]’ 

Additionally, justification of the use of domains from physical activity as an initial framework for classifying the health benefits of performing arts participation is further aided by the clarification of study aims and hypotheses in the revised introduction (p. 4, lines 86-97 and detailed above in the response to point #4). 

11. Page 10, Figure 1: Figure 1 is excellent and a graphical representation of your umbrella review. It would be more helpful to the reader either in text or in the caption to add the specific reason why studies were excluded (Eg. Xx studies excluded because no heart rate data, xx studies excluded because included only in qualitative data etc).

Author Response: Figure 1 has been updated to indicate a breakdown of specific reasons for exclusion. Further, the Figure 1 caption has been revised to direct the reader to the Supplementary Appendix for details regarding excluded reviews and studies (p. 11, lines 251-52): ‘Specific details regarding excluded reviews/studies are contained in the Supplementary Appendix.’ 

12. Page 15, table 2: this is an excellent, extensive list of studies! Were the studies also grouped by style/instrument of dance or music in order to find whether combined together these studies yielded consistent (intensity) of health outcomes? Were the studies also grouped by participation setting (lab/class/performance etc) to see if there were participation setting specific effects?

Author Response: We thank the reviewer for their positive appraisal of our work! Studies were also grouped by style/instrument of dance and music, as noted on p. 17, lines 295-300 of the revised methods:

The effects of music participation were supported by moderate to high quality evidence for 11 individual outcomes, with reported positive effects in 5 domains (auditory; cognitive; immune function/inflammation; self-reported health/wellbeing; social functioning). Modes of performing arts participation associated with the broadest positive health effects were: aerobic dance (4 domains); ballroom dance (4 domains); social dance (4 domains); drumming (3 domains); and Zumba dance (3 domains).

Participation setting was not extracted from included primary studies of the health effects of performing arts participation – this information was only extracted from studies of heart rate responses to performing arts participation. We agree, however, that this is an important consideration for future research, and have noted this on p. 23, lines 400-403: ‘Substantial further research is required to determine the impact of the frequency and duration of performing arts participation on health benefits, as well as the potential additional impact of the setting (e.g. laboratory, classroom, live performance) of performing arts participation on observed benefits.’

13. Page 16 (for some reason line numbers disappear after table 2 � ): Did you have age-specific predictions?

Author Response: A priori review hypotheses have been included in the revised introduction (p. 4, lines 92-94: ‘Performing arts participation is hypothesized to have similar health effects as physical activity due to its intrinsic physical exertion, as well as additional effects related to creative expression and engagement with music.’

14. Page 22, first paragraph: why is the quality of existing studies low? Is this because of low sample sizes? Or missing control groups? More broadly, I think the discussion needs to address the issue of control groups. What control groups were used in primary studies to dissociate the effects of music and/or dance from the effects of other forms of activity?

Author Response: Further details regarding the low quality of existing studies and range of control groups used has been added to p. 22, lines 360-365:

The overall quality of included evidence is generally low (26% (45/173) of outcomes backed by moderate-high quality evidence) due to a predominance of non-randomized and observational vs. randomized controlled trial study designs. Control and comparison groups varied widely across all study types, including no-intervention/waitlist control groups and exercise (various types), cognitive and/or language training and other art participation (e.g. visual art, drama) comparison groups. 

15. Page 22, first paragraph: “validated, sensitive instruments for quantifying performing arts participation” – please clarify what you mean by validated and sensitive instruments, and highlight directs for future research more broadly in the discussion that follow on from your findings.

Author Response: This sentence has been clarified and the directs for future research have been expanded in the revised discussion (p. 23, lines 387-391):

However, the quality of these epidemiologic studies is presently low and specifically limited by the use of a range of bespoke survey instruments with unclear psychometric value to quantify the frequency, timing/duration and type of performing arts participation. Future studies using validated instruments for quantifying performing arts participation are needed.

Reviewer #2: Summary

This is a well-written and clear review of an area lacking important research. This review thoroughly collates the present evidence on the health impacts of music and dance participation and provides a solid foundation for future research.

General points

In particular it is commendable that music and dance are both covered in this review. I think it would be nice to acknowledge in the introduction their common evolutionary origins. The review supports the idea that group dancing and musicking fulfil similar psychological and societal functions https://www.pnas.org/content/112/29/8987.short

https://royalsocietypublishing.org/doi/10.1098/rsbl.2015.0767

https://journals.plos.org/plosone/article?id=10.1371/journal.pone.0180101

https://www.ncbi.nlm.nih.gov/pmc/articles/PMC6092630/

Author Response: We agree that the common evolutionary origins of music and dance are important to acknowledge and have revised the introduction to include this and key suggested references (p. 3, lines 68-70): ‘Within the performing arts, music and dance participation are the two most popular modes of engagement, both involving engagement with music, and proposed to have common evolutionary origins.[7-9]’ 

Specific points

1) Page, 3, first paragraph, a recent systematic review on dance interventions for mental health was published by Millman et al., 2020 here https://onlinelibrary.wiley.com/doi/full/10.1002/cpp.2490

Author Response: While we agree that the Millman review is both relevant to this umbrella review and a valuable recent review, this sentence has been revised to clarify that we are referring specifically to the recent World Health Organization-commissioned scoping review on the health effects of the arts (p. 3, lines 59-61): ‘Participation and receptive engagement in the arts are increasingly recognized as being health promoting, most notably in policy briefs,[1] a recent World Health Organization-commissioned scoping review,[2] and social prescribing initiatives.[3]’

2) Page 20, line 1/2 – “This umbrella review…” and “…evidence demonstrating that performing arts participation is, broadly, a health promoting activity…”

Author Response: We thank the reviewer for these corrections. The suggested changes have been made (p. 21, lines 344-345).

3) Page 21 line 1 – “…greater amounts of high quality studies of specific…”

Author Response: We thank the reviewer for these corrections. The suggested changes have been made (p. 22, line 359).

4) Page 23, line 18 – “Performing arts participation is a broadly healthy promoting activity, with…” or could do the same as what I suggested for Page 20 “… is, broadly, a health promoting…”

Author Response: We thank the reviewer for these corrections. The suggested changes have been made (p. 25, line 432).

---

## [Decision Letter · Decision Letter 1]

26 May 2021

Performing arts as a health resource? An umbrella review of the health impacts of music and dance participation

PONE-D-20-39389R1

Dear Dr. McCrary,

We’re pleased to inform you that your manuscript has been judged scientifically suitable for publication and will be formally accepted for publication once it meets all outstanding technical requirements.

Kind regards,

Emily S. Cross

Academic Editor

PLOS ONE

Reviewers' comments:

Reviewer's Responses to Questions

**Comments to the Author**

1. If the authors have adequately addressed your comments raised in a previous round of review and you feel that this manuscript is now acceptable for publication, you may indicate that here to bypass the “Comments to the Author” section, enter your conflict of interest statement in the “Confidential to Editor” section, and submit your "Accept" recommendation.

Reviewer #1: All comments have been addressed

2. Is the manuscript technically sound, and do the data support the conclusions?

Reviewer #1: Yes

3. Has the statistical analysis been performed appropriately and rigorously? 

Reviewer #1: Yes

4. Have the authors made all data underlying the findings in their manuscript fully available?

Reviewer #1: Yes

5. Is the manuscript presented in an intelligible fashion and written in standard English?

Reviewer #1: Yes

6. Review Comments to the Author

Reviewer #1: All comments have been satisfactorily addressed, and I wish the authors good luck for their publication!

7. PLOS authors have the option to publish the peer review history of their article (what does this mean?). If published, this will include your full peer review and any attached files.

Reviewer #1: **Yes: **Kohinoor M. Darda

---

## [Editor Report · Acceptance letter]

28 May 2021

PONE-D-20-39389R1 

Performing arts as a health resource? An umbrella review of the health impacts of music and dance participation 

Dear Dr. McCrary:

I'm pleased to inform you that your manuscript has been deemed suitable for publication in PLOS ONE. Congratulations! Your manuscript is now with our production department. 

Kind regards, 

on behalf of

Professor Emily S. Cross 

Academic Editor

PLOS ONE